# Motion Deblurring for Single-Pixel Spatial Frequency Domain Imaging

**Mai Dan [1], Meihui Liu [1] and Feng Gao [1,2,*]**

[1] Department of Biomedical Engineering, College of Precision Instrument and Optoelectronics Engineering, Tianjin University, Tianjin 300072, China; danmai@tju.edu.cn (M.D.); lmhui@tju.edu.cn (M.L.)

[2] Tianjin Key Laboratory of Biomedical Detecting Techniques and Instruments, Tianjin 300072, China

[*] Correspondence: gaofeng@tju.edu.cn

**Abstract:** The single-pixel imaging technique is applied to spatial frequency domain imaging (SFDI) to bring significant performance advantages in band extension and sensitivity enhancement. However, the large number of samplings required can cause severe quality degradations in the measured image when imaging a moving target. This work presents a novel method of motion deblurring for single-pixel SFDI. In this method, the Fourier coefficients of the reflected image are measured by the Fourier single-pixel imaging technique. On this basis, a motion-degradation-model-based compensation, which is derived by the phase-shift and frequency-shift properties of Fourier transform, is adopted to eliminate the effects of target displacements on the measurements. The target displacements required in the method are obtained using a fast motion estimation approach. A series of numerical and experimental validations show that the proposed method can effectively deblur the moving targets and accordingly improves the accuracy of the extracted optical properties, rendering it a potentially powerful way of broadening the clinical application of single-pixel SFDI.

**Keywords:** spatial frequency domain imaging; motion deblurring; single-pixel imaging

## 1. Introduction

Spatial frequency domain imaging (SFDI) works by projecting the sinusoidal structured light onto the tissue and measuring the light reflected from the tissue surface [1,2]. It enables fast, non-invasive and wide field mapping of the tissue optical properties, and has been applied to many clinical applications [3–12]. Recently, since the single-pixel imaging (SPI) technique has brought significant performance advantages in band extension and sensitivity enhancement [13,14], single-pixel SFDI is receiving extensive attention [15–20].

Despite its combined merits of both SPI and SFDI, single-pixel SFDI has been restricted to low frame rates because it requires hundreds of measurements to obtain a single image. This adversity can cause severe quality degradations of the measured reflected images when imaging a moving target, such as scenarios of the in vivo imaging of animals in motion or clinical interrogation of patients for breathing and heartbeat, and accordingly reduces the accuracy of the extracted optical properties. Despite the many techniques proposed for motion deblurring in the image restoration regime, they fail in direct translation to single-pixel SFDI because of its essentially different imaging mechanism from the traditional photography.

Many works have been devoted to improving the dynamic performance of SPI. For instance, the high-speed illumination module [21] and digital micromirror device (DMD) [22,23] were exploited to achieve a high frame rate. However, these dynamically enhancing methods normally require high illuminating power to ensure a reasonable signal-to-noise ratio (SNR) at a high refresh rate, which, therefore, can cause harm to tissues when applied to single-pixel SFDI. Other methods obtain the high-quality image by successively

transforming the illumination patterns according to the target locations or superimposing a sequence of under-sample images, with each image shifted with the target displacement [24–30]. The motion information required in these methods is estimated by the optimization algorithms or using the cross-correlation or the low-order moments of the under-sample images. However, these methods require the image to possess a uniform background because the target motion is regarded as the translation of the entire image. This requirement prevents these methods from solving the motion blurring in single-pixel SFDI, as the images in single-pixel SFDI have the sinusoidal fringes in the background that remain stationary when the target moves.

In this work, we present a novel method of motion deblurring for single-pixel SFDI. The method adopts a Fourier single-pixel imaging (FSI) technique to acquire the Fourier coefficients of the reflected image. On this basis, a motion-degradation-model-based compensation, which is derived by the phase-shift and frequency-shift properties of Fourier transform, is adopted to eliminate the effects of target displacements on the measurements. Thereafter, the deblurred image is reconstructed from the processed data. The target displacements required in the method are obtained using a fast motion estimation approach. The novelty of this work is to derive the motion degradation model for single-pixel SFDI that describes the effects of target displacements on both the planar (DC) and spatially modulation (AC) components of the image. The compensation of the measurements based on this model can deblur the target while retaining the background sinusoidal fringes. The method enables the fast deblurring of targets with varying speeds and directions. We performed a series of numerical simulations and phantom experiments to validate the proposed method. The numerical results show that the method can dynamically locate and deblur the target in the reflected image, and the phantom experiments further demonstrate the method to significantly improve the accuracy of the subsequently extracted optical properties.

## 2. Materials and Methods

The motion degradation model for single-pixel SFDI is first introduced in this section. The motion estimation approach is then described that obtains the required target displacements. In addition, a circular sampling strategy that can maximize the sampling efficiency and a dynamic reconstruction scheme that can accelerate the frame rate of single-pixel SFDI are adopted, respectively.

### 2.1. Motion Degradation Model for Single-Pixel SFDI

In single-pixel SFDI, the reflected image is reconstructed from a sequence of sampling patterns and their corresponding measurement values. When the target moves, the reflected image at the $i_{th}$ measurement is

$$I_i(x,y) = M_{DC}(x + \triangle x_i, y + \triangle y_i) + M_{AC}(x + \triangle x_i, y + \triangle y_i) \cdot \cos(2\pi f \cdot x + \varphi_0), \quad (1)$$

where $M_{DC}(x,y)$ and $M_{AC}(x,y)$ are the DC and AC amplitude images; $\triangle x_i$ and $\triangle y_i$ are the target displacements in the X and Y directions at the $i_{th}$ measurement (taking the image at the first measurement as the true image), respectively; $f$ is the spatial frequency of the sinusoidal illumination; and $\varphi_0$ is the initial phase. The target moves between measurements, but the sinusoidal illumination in the imaging field of view (FOV) remains stationary during the imaging process. This requires that the motion deblurring only works on the moving target while retaining the background sinusoidal fringes.

In this method, the motion degradation model is derived by the two properties of the Fourier transform that the sinusoidal modulation of an image in the spatial domain results in a frequency shift in the Fourier domain (frequency-shift property), and the translation of an image in the spatial domain results in a phase shift in the Fourier domain (phase-shift property). By transforming the reflected image into the Fourier domain, the sinusoidal fringes can be simplified as an impulse component that shifts the spectral center of the

AC image to the frequency of $f$. The Fourier spectrum of the reflected image is therefore separated into the DC and AC components. Then, according to the phase-shift property, the effects of the target displacements on the DC and AC components can be eliminated by the phase-shift compensation.

To facilitate the processing in the Fourier domain, the FSI that acquires the Fourier coefficients of the image is adopted. Within the short time of a single measurement, the target can be treated as relatively static. The moving process is accordingly discretized into M periods, each corresponding to a measurement. The Fourier coefficient acquired at the $i_{th}$ measurement satisfies the phase-shift and frequency-shift properties, as,

$$F'_\upsilon(u,v) = F_\upsilon(u,v) \cdot e^{-j2\pi(\triangle x_i \cdot \frac{u-u_0}{N} + \triangle y_i \cdot \frac{v}{N})} + N(u,v),\ u \geq 0, \tag{2}$$

where $F_\upsilon(u,v)$ and $F'_\upsilon(u,v)$ are the DC or AC Fourier components of the true and blurry images, respectively; $\{DC, AC\} \in \upsilon$; $N(u,v)$ is the noise spectrum; $u$ and $v$ are the spatial frequencies of the Fourier coefficients; $u_0$ is the center frequency of the sinusoidal illumination, where $u_0 = 0$ for the DC image; and N is the pixel resolution of the image. Here, it is assumed that the target moves within the FOV during the entire imaging procedure. Because the Fourier spectrum of a natural image is conjugate symmetric, only the Fourier coefficients for $u \geq 0$ are required to measure. The coefficients for $u < 0$ are obtained using the conjugate symmetry.

The imaginary term in Equation (2) is denoted as the degradation function for the DC or AC component,

$$H_\upsilon(u,v) = e^{-j2\pi(\triangle x_i \cdot \frac{u-u_0}{N} + \triangle y_i \cdot \frac{v}{N})},\ u \geq 0, \tag{3}$$

The degradation function for the AC component has a frequency shift of $u_0$ in the X direction due to the sinusoidal modulation. The phase shifts caused by the target displacements can be compensated by inverse filtering based on the degradation functions,

$$\hat{F}_\upsilon(u,v) = \frac{F'_\upsilon(u,v)}{H_\upsilon(u,v)}, \tag{4}$$

Because the DC and AC components are separated and concentrated in their respective central frequencies, the phase-shift compensation works on them independently. Thereafter, two deblurred images $\hat{I}_{DC}(x,y)$ and $\hat{I}_{AC}(x,y)$ are obtained by performing inverse Fourier transforms to $\hat{F}_\upsilon(u,v)$. The DC and AC amplitude images are then obtained using the single-snapshot demodulation [31]. In this work, only the low-frequency Fourier coefficients (dominated by signals) of the reflected image are measured, which prevents the inverse filtering from amplifying the noise spectrum. When high-frequency Fourier coefficients are measured, wiener filtering can be adopted as the alternative to inverse filtering to inhibit the noise amplifications.

### 2.2. Motion Estimation

The target displacement at each measurement is obtained by the fast motion estimation approach. The approach calculates the displacements from the repeated measurements of the same Fourier coefficients,

$$u \cdot \triangle x_K + v \cdot \triangle y_K = -\frac{N}{2\pi} \cdot \arg\{\frac{Y_{i+K}}{Y_i}\}, \tag{5}$$

where $Y_i$ and $Y_{i+K}$ are the detected values of the same Fourier basis pattern at the $i_{th}$ and $(i+K)_{th}$ measurements, respectively; $(\triangle x_K, \triangle y_K)$ is the target displacement between the two measurements; and $\arg\{\cdot\}$ denotes argument operation. Using two pairs

of Fourier coefficients, a system of equations can be established for solving $(\triangle x_K, \triangle y_K)$. A similar approach was employed for motion detection and tracking in previous works [32–34]. For simplicity, the Fourier coefficients of $F(f_x, 0)$ and $F(0, f_y)$, where $f_x = f_y = 2/N$, are used to calculate $\triangle x_K$ and $\triangle y_K$, respectively. The two Fourier coefficients are adjacent to the zero frequency, which can reduce the interference of the AC component to the displacement estimation. The corresponding Fourier basis patterns, $P_x$ and $P_y$, are measured between each $K$ samples to dynamically track and fit the target locations. The displacement at each measurement is then obtained from a nonlinear interpolation of the results. The schematic diagram of the sampling patterns for motion estimation and image reconstruction is shown in Figure 1.

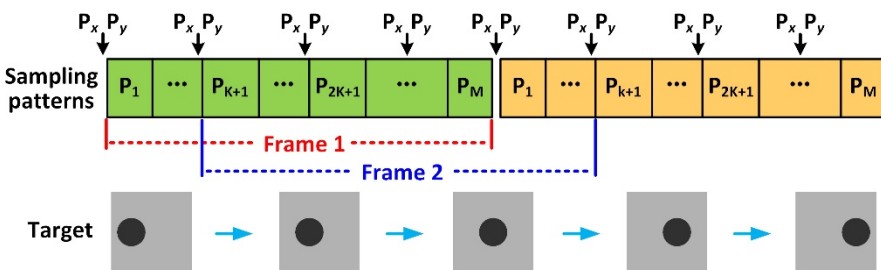

**Figure 1.** Schematic diagram of the sampling patterns for motion estimation and dynamic reconstruction.

To accelerate the imaging frame rate, a dynamic reconstruction scheme using the temporally reciprocating and overlappingly framing of sampling patterns is adopted, as shown in Figure 1. In the scheme, the sampling patterns are measured sequentially in a predetermined priority. Any M continuous measurements constitute an imaging window for reconstructing a frame that reflects the target information during this period. When reconstructing a new frame, only the K new measurements are updated, while the other measurements share the data from the previous frame. The dynamic reconstruction scheme is capable of obtaining more frames from the measurements compared to the conventional SPI.

### 2.3. Sampling Strategy for Single-Pixel SFDI

FSI inherently has a high sampling efficiency because the energy of natural images is normally concentrated at low frequencies in the Fourier domain. The sampling strategy along a circular path is the most effective for a natural image [35]. In single-pixel SFDI, the energy of the reflected images is concentrated at the zero frequency (DC component) and the modulation frequency (AC component), as shown in Figure 2a. To maximize the sampling efficiency for single-pixel SFDI, a modified circular path is employed, as shown in Figure 2b.

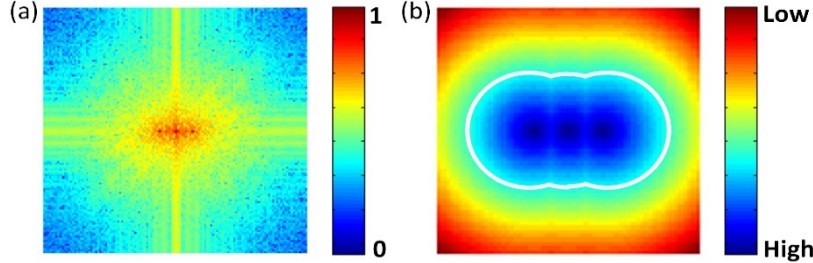

**Figure 2.** Sampling strategy for single-pixel SFDI. (**a**) Normalized Fourier spectrum of a reflected image, (**b**) modified circular path (white curves) and measuring priority of sampling patterns.

Because SFDI uses both the DC and AC diffuse reflectances to extract the optical properties, it is essential to design the measuring priority of the Fourier coefficients to ensure the synchronous updates of the DC and AC components in the dynamic reconstructions. The measuring priority is determined according to the minimum Euclidean distance of each Fourier coefficient from the two center frequencies, zero frequency and $f$. The Fourier coefficients with smaller distances normally contribute more to the recovery of the DC or AC component, and therefore have higher measuring priority, as shown in Figure 2b.

## 3. Results

We performed a series of numerical and experimental validations to evaluate the performance of the proposed method. The results were compared with those obtained using the conventional single-pixel SFDI. Those results show that the proposed method can effectively deblur the reflected images and accordingly improves the accuracy of the subsequently extracted optical properties.

### 3.1. Simulation Validations

In the simulation validations, a letter "E" is used as the target, modulated by the spatial frequency of 0.2 mm$^{-1}$, as shown in Figure 3a. The pixel resolution of the image is 128 × 128. The target in piecewise uniform motion along X and Y directions and in circular motion with a constant angular acceleration are simulated, respectively. 500 sampling patterns are used to reconstruct the reflected image, with $P_x$ and $P_y$ measured every 10 samplings ($K = 10$) to estimate target displacements. A single frame is measured in the piecewise uniform motion simulation, while two complete frames are measured in the circular motion simulation. Then, 30 dB Gaussian noise is added to the simulated data. The structural similarity index measure (SSIM) is used to evaluate the effects of the method.

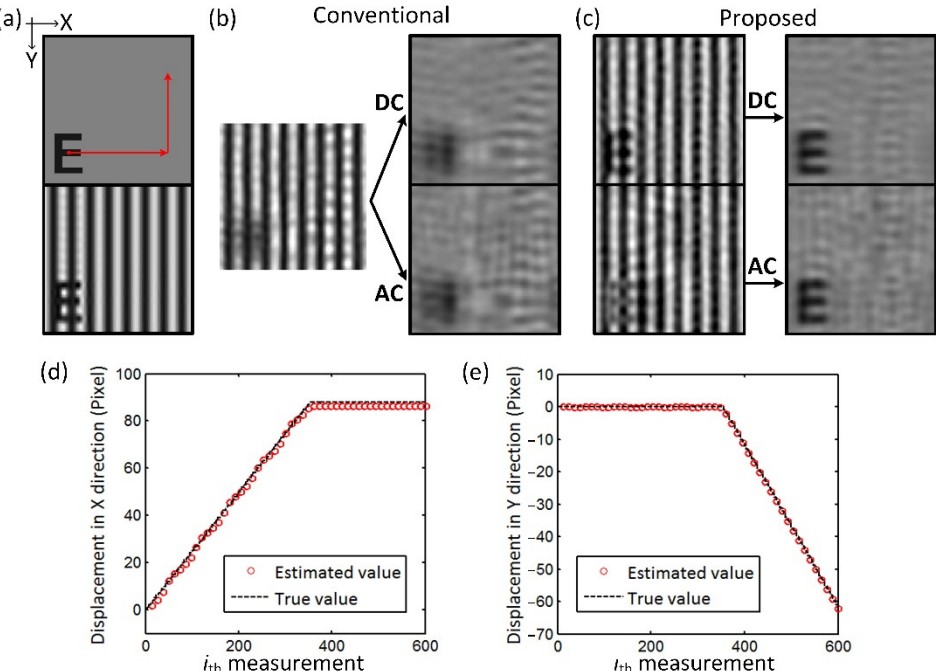

**Figure 3.** Simulation results of the target in piecewise uniform motion. (**a**) True target images. Red arrows show the moving path and directions of the target. Results obtained using (**b**) conventional single-pixel SFDI and (**c**) the proposed method. Target displacements in (**d**) X and (**e**) Y directions, calculated from motion estimation.

The results of the uniform motion simulation are shown in Figure 3. The target successively moves 90 pixels in the X direction and 60 pixels in the Y direction within 600 measurements. Figure 3b,c show the results obtained using the proposed method and conventional single-pixel SFDI, respectively. In the conventional single-pixel SFDI method, the reflected image reconstructed is severely degraded due to the target motion in the measurements. The targets in the demodulated DC and AC images are completely illegible. The proposed method restores the degraded image for both the DC and AC components and obtains the demodulated images with clear details. The SSIM of the DC and AC images between Figure 3c and Figure 3a are 0.91 and 0.86, compared to the values of 0.53 and 0.49 between Figure 3b and Figure 3a, respectively. Figure 3d shows the target displacements in the X and Y directions estimated from the measurement values. The sinusoidal modulation reduces the accuracy of estimated displacement in the X direction to some extent, but the results are within the error of <2%.

The results of circular motion simulation are shown in Figure 4. The target moves clockwise in a circle with a constant angular acceleration, moving one turn in 1200 measurements, as shown in Figure 4a. A total of 50 frames were obtained using the dynamic reconstruction scheme. Figure 4b,c show the results of Frame #30 reconstructed by the conventional single-pixel SFDI and the proposed method, respectively. The mean SSIM of DC and AC images of all frames is increased from 0.58 and 0.55 to 0.91 and 0.86 after the restorations. The estimated displacements in the X and Y directions are also within the error of 2% of the true values. The angular velocities are also calculated from the estimated displacements and are fitted using linear regression, as shown in Figure 4d. The angular acceleration (i.e., the slope of the fitting line) is within an error of <0.2%, proving the capability of the proposed method to dynamically locate the moving target with varying speeds and directions.

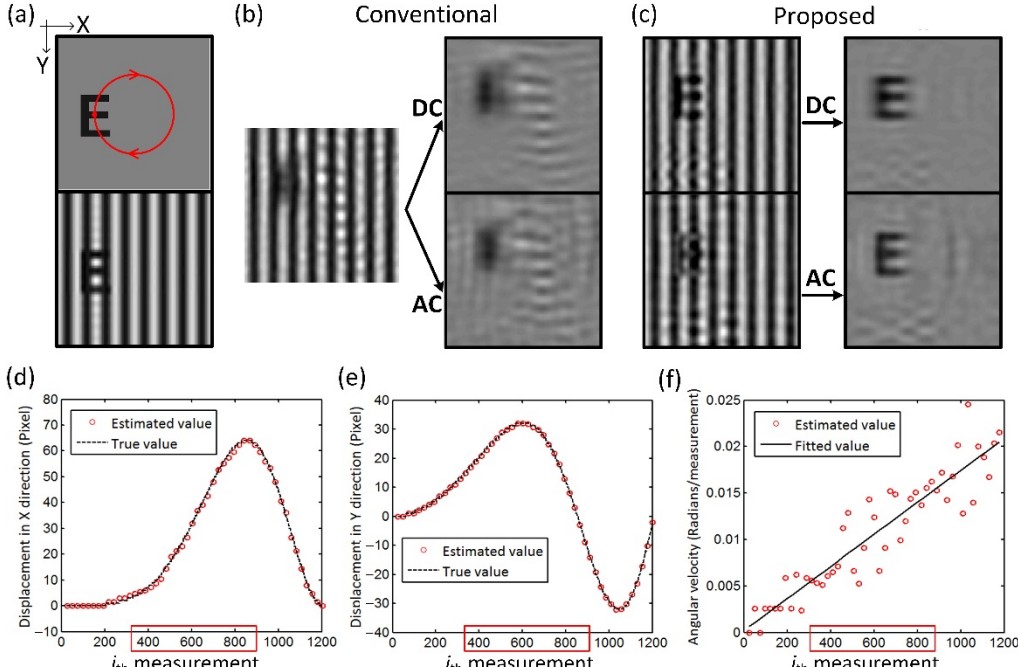

**Figure 4.** Simulation results of the target in circular motion. (**a**) True target images. Red line and arrows show the moving path and directions of the target. Results (Frame #30) obtained using (**b**) conventional single-pixel SFDI and (**c**) the proposed method. Target displacements in (**d**) X and (**e**) Y directions and (**f**) angular velocities, calculated from motion estimation. The red windows indicate the measurements used for reconstructing Frame #30.

*3.2. Experimental Validations*

The experimental validations were performed using a custom-developed single-pixel SFDI system [17]. In the system, two DMDs are used as the spatial light modulator, one generating the sinusoidal illumination pattern for projection, and the other generating the Fourier basis patterns for spatially encoding the reflected images. The light signal is detected by a high-sensitivity photomultiplier tube. The DMD refresh rate used in the experiment is 40 Hz. The pixel resolution of the image is 128 × 128, over the FOV of 40 × 40 mm². The spatial frequency of the sinusoidal illumination is 0.2 mm⁻¹. The number of samplings for image reconstruction is 132, and the $P_x$ and $P_y$ are measured every 10 samplings ($K = 10$) for motion estimation. Then, 3-step phase-shifting FSI is adopted to acquire the complex-valued Fourier coefficients, resulting in a total of 474 measurements to obtain a deblurred image.

A tissue-like phantom is used in the experiment. The phantom was fabricated using India ink as the main absorber, Intralipid-20% as the main scatterer, and was cured using agar. The phantom is 70 cm × 70 cm × 30 cm in size with the background absorption coefficient ($\mu_a$) of 0.01 mm⁻¹ and the reduced scattering coefficient ($\mu_s'$) of 1.2 mm⁻¹. A cylindrical target with the diameter of 1 cm and the depth of 0.5 cm is in the middle of the phantom. The $\mu_a$ and $\mu_s'$ of the target are 0.04 mm⁻¹ and 1.2 mm⁻¹, respectively. In the experiment, the phantom moves in different speeds within the FOV. The speed is controlled by manually operating a linear translation stage. Since manual control is hard to ensure a constant speed, we use the average speed as the true value, which is calculated according to the total measurement time and the total displacement of the target. The average speeds of the phantom are approximately $v = 0$, 0.5, 1, 1.5, 2 and 2.5 mm/s, respectively. The maximum validated speed is 2.5 mm/s, as the higher speed will force the target to move out of the FOV.

The reflected images of the phantom at different speeds were obtained using the conventional reconstruction method and the proposed method, respectively. The optical properties were then extracted from the reflectance data. Figure 5 shows the optical property results of the static phantom and the phantom at different speeds, respectively. When the phantom moves, the absorption maps obtained by the conventional single-pixel SFDI have severe artifacts, resulting in significant reductions in the quantitative accuracy. The scattering maps do not have the motion artifacts, as the target has the same reduced scattering coefficient as the background. In contrast, the proposed method effectively deblurs the target and improves the quantitative accuracy. The estimated speeds of the phantom are within the error of 8%. The SSIM and root-mean-square error (RMSE) of the results are calculated. The results of the static phantom are used as the reference. The proposed method significantly improves the SSIM and RMSE of the results at all speeds compared with the conventional method, as shown in Table 1. The results indicate that the proposed method has a strong ability to image a moving target.

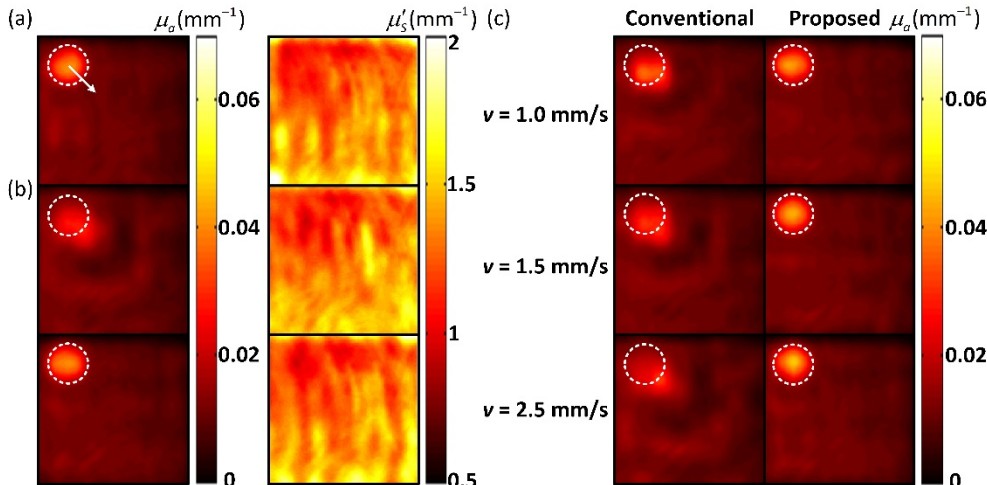

**Figure 5.** Experimental results of moving phantom. Absorption and scattering property maps of (**a**) the static phantom and (**b**) the phantom at $v = 2$ mm/s obtained using the conventional single-pixel SFDI (top) and the proposed method (bottom), respectively. (**c**) Absorption maps of the phantom at different speeds. The dashed circles and white arrow indicate the true location and moving direction of the target, respectively.

**Table 1.** SSIM and RMSE of the experimental results.

| Speed (mm/s) | RMSE (%) | | SSIM | |
|---|---|---|---|---|
| | Conventional | Proposed | Conventional | Proposed |
| $v = 0.5$ | 0.19 | 0.12 | 0.9960 | 0.9984 |
| $v = 1.0$ | 0.27 | 0.13 | 0.9926 | 0.9982 |
| $v = 1.5$ | 0.31 | 0.14 | 0.9902 | 0.9979 |
| $v = 2.0$ | 0.34 | 0.15 | 0.9878 | 0.9975 |
| $v = 2.5$ | 0.38 | 0.15 | 0.9846 | 0.9975 |

## 4. Discussion

Despite its success, there are limitations of the method. The motion degradation model derived by the phase-shift property works well in dealing with the target translation within the FOV, but for other motions, such as target rotation or target moving out of the FOV, the model requires to be further improved. The current work is limited to process a single target. If multiple targets move within the FOV, the degradation model will be ineffective. Our future work will extend the method to multi-target motion deblurring. In addition, the method can be less effective when the spatial frequency used is low, because the phases of the DC and AC components can overlap with each other to hinder the phase-shift compensation. The low-frequency modulation will also reduce the efficacy of single-snapshot demodulation. We find that when the spatial frequency used is less than 0.1 mm$^{-1}$, the performance of the method is greatly reduced. In practice, the typically used spatial frequency is between 0.1–0.2 mm$^{-1}$, so the method is applicable to most scenarios.

The motion estimation is important for image deblurring. But the accurate tracking of the target requires more frequently measuring two additional Fourier coefficients, which can further increase the number of samples for reconstructing the image. For the uniform motion and the circular motion validated in the numerical and phantom experiments, K value between 10 and 20 can give consideration to both imaging speed and quality. The number of measurements is only increased by less than a fifth. However, if the target moves completely randomly (its trajectory is hard to fit), a very small K value will be required, which could multiply the total number of measurements. In fact, the Fourier coefficients for image reconstruction can also be used for motion estimation. In the dynamic reconstruction, each updated Fourier coefficient has the phase-shift relation with

its previously measured value, which can be used to calculate the target displacement between these two measurements. At the beginning stage of imaging, the errors are inevitable because only the total displacement within each frame is obtained, and the complex frequency components of the Fourier coefficients can further increase the difficulty and error in the calculation. However, as the displacements are updated constantly during imaging, the target locations are able to be tracked and fitted, thus improving the results.

Similar motion estimation approaches using Fourier phase shifts have been reported in previous works. However, these works aimed to achieve an image-free tracking of moving target. Their ultimate goal was to obtain the location of the target, rather than to obtain the deblurred image. In these methods, Fourier phase shifts are only used to estimate target displacements. In contrast, the novelty of our work is to derive the motion degradation model for image deblurring. Our goal is to obtain a high-quality image of a moving target in single-pixel SFDI. Because the existing deblurring methods for SPI cannot handle the modulation background in the SFDI image and no motion deblurring method for single-pixel SFDI is reported, this study is not compared with other suitable methods. However, the comparison with the traditional single-pixel SFDI method can prove the effectiveness of the proposed method.

The dynamic reconstruction strategy, while improving the speed of single-pixel SFDI, requires a trade-off between the frame rate and information redundancy as the measurement data is shared between frames. A more reasonable way to improve the imaging speed is to increase the DMD refresh rate. However, the grayscale sampling patterns required by FSI technique impose a limitation on this implementation as DMD generates grayscale patterns at a low refreshing rate. In this work, the DMD used supports a refresh rate of 120 Hz for grayscale modulation, but up to 4 kHz for binary modulation. A recent work proposed to binarize the grayscale patterns by the Floyd–Steinberg dithering method and demonstrated this strategy to enable high-speed FSI on a DMD-based imaging system [36].

Although this method is targeted to solve the motion blurring for single-pixel SFDI, it has other application areas. The modulation strategy is not only used in SFDI, but also in other imaging modalities, such as structured illumination microscopy and structured light profilometry [37,38]. The realization of these imaging modalities based on SPI technique is receiving wide attention. This method is expected to provide an idea for solving the motion blur in these studies. In addition, our method is also applicable to ordinary scenarios without modulation illumination. In comparison with the previously presented motion deblurring schemes, our proposed method requires no measurement of multiple under-sampled images for motion estimation or transforming of the illumination patterns for image reconstruction. It enables faster motion estimation and image reconstruction, and therefore has better real-time imaging capability.

## 5. Conclusions

In conclusion, this work presents a novel method of motion deblurring for single-pixel SFDI. The method adopts the FSI technique to measure the Fourier coefficients of the reflected images and uses the model-based phase-shift compensation to eliminate to effects of target motion on the measurements. Experimental results prove that the method can effectively deblur the moving target in the measured reflected image, and accordingly improves the accuracy in the subsequently extracted optical properties. The scenes imaged in the experiments are with uniform backgrounds, but the method also works with non-uniform backgrounds. Because the Fourier spectrum of the scene is the superposition between the Fourier spectrum of the target image and the background image, background subtraction can be adopted before deblurring the target image.

**Author Contributions:** Methodology, M.D.; validation, M.D. and M.L.; writing review and editing, M.D.; supervision, F.G. All authors have read and agreed to the published version of the manuscript.

**Funding:** This research was funded by National Natural Science Foundation of China (81871393, 62075156).

**Institutional Review Board Statement:** Not applicable.

**Informed Consent Statement:** Not applicable.

**Data Availability Statement:** The data presented in this study are available on request from the corresponding author. The data are not publicly available due to laboratory guidelines.

**Conflicts of Interest:** The authors declare no conflict of interest.

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
