# Peer review of "Motion Deblurring for Single-Pixel Spatial Frequency Domain Imaging"

_applsci, doi:10.3390/app12157402_

Round 1

Reviewer 1 Report

The paper is well written. There are some minor remarks that can improve its quality before the publication.

- Charts in Fig. 3 and 4 can be easier readable.

- in Discussion section it'd nice to have a comparison to other state-of-the-art articles, not only the comparison Conventional/Proposed.

Author Response

Thanks very much for your comments and suggestions. We have studied the comments carefully and have made corrections. We appreciate for reviewers’ warm work earnestly, and hope that the corrections will meet with approval.

point 1: Charts in Fig. 3 and 4 can be easier readable.

Response 1: Thanks for your comments. We have revised Fig. 3 and 4 to make them easier readable.

point 2: in Discussion section it'd nice to have a comparison to other state-of-the-art articles, not only the comparison Conventional/Proposed.

Response 2: We have compared the study with the state-of-the-art methods in the discussion section. The discussions in detail are provided in the revised manuscript.

Reviewer 2 Report

I have read the article titled ` Motion deblurring for single-pixel spatial frequency domain  imaging`.

This work, as its title implies is a research article investigating theoretical and experimental work in a imaging system.

The article is in the field of fundamental imaging science.  In this
aspect it can be considered as  a nice applied research
article.

The article reads well and is interesting, with nice and applicable results.

 I therefore would in principle agree that the manuscript be published
in Applied Sciences , after the many issues raised in this review are all responded
to. Especially the ones concentrating on the experiment itself.

Please note that Lenz and colleagues recent work on “Single-pixel spatial frequency domain imaging with integrating detection” should be incorporated in references.

This paper presents a novel method of motion deblurring for single-pixel SFDI. In the method, the Fourier coefficients of the reflected image are  measured by Fourier single-pixel imaging technique. On this basis, a motion degradation model  based compensation, which is derived by the phase-shift and frequency-shift properties of Fourier  transform, is adopted to eliminate the effects of target displacements on the measurements. The  target displacements required in the method are obtained using a fast motion estimation approach.  A series of numerical and experimental validations show that the proposed method can effectively  deblur the moving targets and accordingly improves the accuracy of the extracted optical properties, rendering it a potentially powerful way of broadening the clinical application of SP-SFDI.

This can indeed have relevant and  interesting in many imaging problems with moving targets.

Please proofread by spell check, correcting typos and trivial grammatical mistakes.

Many long sentences are difficult to follow, please clarify and rewrite.

The following sentence is not clear: on lines 131 and 132

Why is 0.0625 chosen, relevance etc?

Is the case mentioned in reference 33 on line 152 actually  relevant for this experiment, justify?

In Figure 3, the DC and AC components are exactly the same, explain?It looks like a mistake, correct, double check?

Please note that a modulation strategy was used in a laser imaging system in order to reduce the speckle, ie increase SNR of image, as in the case

Yilmazlar “Implementation of a current drive modulator for effective speckle suppression in a laser projection system”, IEEE Photonics Journal

It would be interesting to compare a moving target deblurring imaging strategy with the mentioned one.

The article would therefore may have other application areas. It would add to other nonclinical applications and would broaden readership of article.

In Figure 5 and in the simulations given in Table, how were the target velocities determined?

Other Points:

Explain clearly the novelty of the work. This should be given in the abstract and main body. Not just as an extension to previous study but underline originality of this present manuscript..

Some Figure  is not properly visible. Prepare a clearer depiction if possible, increase resolution.

What are the limitations of the study? Elaborate.

Compare  with state of the art or other relevant methods?

Add relevant mentioned references above and other..

Please elaborate the outlook and comment more on real life effects on practical medical and other imaging.

I  would therefore suggest its recommendation to be published in journal, once all
the issues raised in this review report are responded to.

Author Response

Thanks very much for your comments and suggestions. We have studied the comments carefully and have made corrections. We appreciate for reviewers’ warm work earnestly, and hope that the corrections will meet with approval.

Point 1: Please note that Lenz and colleagues recent work on “Single-pixel spatial frequency domain imaging with integrating detection” should be incorporated in references. Please proof read by spell check, correcting typos and trivial grammatical mistakes. Many long sentences are difficult to follow, please clarify and rewrite. The following sentence is not clear: on lines 131 and 132 Why is 0.0625 chosen, relevance etc? Is the case mentioned in reference 33 on line 152 actually relevant for this experiment, justify? In Figure 3, the DC and AC components are exactly the same, explain? It looks like a mistake, correct, double check? In Figure 5 and in the simulations given in Table, how were the target velocities determined?

Response 1: Thanks for your comments. We have checked the work of Lenz and et al., and have incorporated it in references (ref 20). We have carefully checked sentences and corrected typos and grammatical mistakes. We have carefully checked all the results and figures. The DC and AC components in Figure 3 look similar, but they are not the same (their SSIM values are different). The difference of the two images can be distinguished from their backgrounds.

As for the chosen spatial frequency on lines 131 and 132, we apologize that the value of 0.0625 is a typo. We have corrected it in the manuscript (lines 134-137). We choose two Fourier coefficients adjacent to the zero frequency, which are denoted by  and , respectively, with . The use of these two Fourier coefficients can reduce the interference of the AC component to the displacement estimation. In the experiment, the two Fourier coefficients correspond to the actual spatial frequencies of 0.0125 mm-1 in the x and y directions, respectively.

The circular sampling strategy adopted in our work is inspired by reference 33. The work in reference 33 compares three sampling strategies and demonstrates the circular sampling path the most effective for Fourier single-pixel imaging. We think it should be referenced.

In the experiment, the speed of the phantom was controlled by manually operating the linear translation stage. Since manual control is hard to ensure a constant speed, we use the average speed as the true value. The average speed is calculated from the total measurement time and the total displacement of the target. We did a lot of practice before the measurement to try to keep the target speed as close to uniform as possible. The descriptions are included in the manuscript (lines 245-248).

Point 2: Please note that a modulation strategy was used in a laser imaging system in order to reduce the speckle, i.e. increase SNR of image, as in the case Yilmazlar “Implementation of a current drive modulator for effective speckle suppression in a laser projection system”, IEEE Photonics Journal. It would be interesting to compare a moving target deblurring imaging strategy with the mentioned one. The article would therefore may have other application areas. It would add to other nonclinical applications and would broaden readership of article.

Response 2: Thanks for your comments. We have checked the work mentioned above. The modulation strategy used in the mentioned work is a temporary modulation. The temporal modulation strategy can improve SNR of measurement and enable multi-wavelength detection. In fact, our custom-developed single-pixel SFDI system used a similar temporal modulation strategy, i.e. the lock-in photon counting technique (see ref 17 for details). But the method in this study is developed to deal with the spatial modulation in the SFDI image. The spatial modulation in SFDI is essentially different from the temporal modulation strategy, so it's not appropriate to compare our method to the work mentioned above. But the spatial modulation strategy is also used in other imaging modalities, such as structured illumination microscopy and structured light profilometry. The realization of these imaging modalities based on single-pixel imaging technique is receiving wide attention. Our method is expected to provide an idea for solving the target motion in these studies. The discussions in detail are provided in the revised manuscript.

Other Points:

  1. Explain clearly the novelty of the work. This should be given in the abstract and main body. Not just as an extension to previous study but underline originality of this present manuscript.
  2. Some Figure is not properly visible. Prepare a clearer depiction if possible, increase resolution.
  3. What are the limitations of the study? Elaborate.
  4. Compare with state of the art or other relevant methods?
  5. Add relevant mentioned references above and other.
  6. Please elaborate the outlook and comment more on real life effects on practical medical and other imaging.

Responses: Thanks for your comments. We have increased the resolution of figures, and we have highlighted the novelty of this work in the main body (lines 61-65). The novelty of this work is to derive the motion degradation model for single-pixel SFDI that describes the effects of target displacements on both the DC and AC components of the image. The compensation of the measurements based on this model can deblur the target while retaining the background sinusoidal fringes. Detailed responses to these points are provided in the discussion section of the revised manuscript.

Reviewer 3 Report

The basic method reported in this paper, use of Fourier phase shifts  has already been published as  the authors admit on page 3 line 129:. "Similar approach has been employed for motion detection and tracking in previous works [31,32].  It has been extended recently to multi-particle tracking 

Zhang, J.; Hu, T.; Shao, X.;Xiao, M.; Rong, Y.; Xiao, Z. "Multi-Target Tracking Using Windowed Fourier Single-Pixel Imaging". Sensors 2021, 21, 7934.
https://doi.org/10.3390/s21237934

The present paper is well written, but the authors have not explained clearly what  the new contribution(s) are. Its usefulness for biomedical imaging is also doubtful, since it appears that the method only works with uniform backgrounds.

The comparisons to the conventional method are not fair since the so called conventional method does not account for target motion. Better to compare to the methods of ref 31 and/or the Sensors paper referenced above.

If the authors can explain clearly which parts are novel, and make fair comparisons, then the paper could be considered for publication.

Author Response

Thanks very much for your comments and suggestions. We have studied the comments carefully and have made corrections. We appreciate for reviewers’ warm work earnestly, and hope that the corrections will meet with approval.

Point :The basic method reported in this paper, use of Fourier phase shifts has already been published as the authors admit on page 3 line 129: "Similar approach has been employed for motion detection and tracking in previous works [31,32]. It has been extended recently to multi-particle tracking: Zhang, J.; Hu, T.; Shao, X.;Xiao, M.; Rong, Y.; Xiao, Z. "Multi-Target Tracking Using Windowed Fourier Single-Pixel Imaging". Sensors 2021, 21, 7934.

The present paper is well written, but the authors have not explained clearly what the new contribution(s) are. Its usefulness for biomedical imaging is also doubtful, since it appears that the method only works with uniform backgrounds. The comparisons to the conventional method are not fair since the so called conventional method does not account for target motion. Better to compare to the methods of ref 31 and/or the Sensors paper referenced above. If the authors can explain clearly which parts are novel, and make fair comparisons, then the paper could be considered for publication.

Response: We have checked the work published on Sensors. This work reports an image-free method for tracking multiple targets. Same as ref 31 and 32, the ultimate goal of these works is to obtain the location of the targets, rather than to obtain a deblurred image. In these methods, Fourier phase shifts are only used to estimate target displacements. In contrast, our goal is to deblur the measured image in single-pixel SFDI. Motion estimation is only one step required for image deblurring. The novelty of our work is to derive the motion degradation model for image deblurring in single-pixel SFDI. The model describes the effects of target displacements on both the DC and AC components of the image. The phase-shift compensation of the measurements based on this model can deblur the target while retaining the background sinusoidal fringes. In Comparison with the existing deblurring methods for SPI, our method can handle the modulation background in the image. The discussions in detail are provided in the revised manuscript.

In terms of the background issue, the scenes imaged in the experiments are with uniform backgrounds, but the method also works with non-uniform backgrounds. Because the Fourier spectrum of the scene is the superposition between the Fourier spectrum of the target image and the background image, background subtraction can be adopted before deblurring the target image. Because the existing deblurring methods for SPI cannot handle the modulation background in SFDI images and no motion deblurring method for single-pixel SFDI is reported, this study is not compared with other methods. But the comparison with the traditional single-pixel SFDI method can prove the effectiveness of the proposed method. The discussions in detail are provided in the revised manuscript.

Round 2

Reviewer 3 Report

The revised paper makes a much more convincing case for publication, and explains the authors' contributions well. I appreciate the fact that shortcomings of the method are included as an impetus to further research.